# LC-MS/MS Profiling of Post-Transcriptional Modifications in Ginseng tRNA Purified by a Polysaccharase-Aided Extraction Method

**DOI:** 10.3390/biom10040621

**Published:** 2020-04-17

**Authors:** Tongmeng Yan, Kua Hu, Fei Ren, Zhihong Jiang

**Affiliations:** State Key Laboratory of Quality Research in Chinese Medicine, Macau University of Science and Technology, Macau 999078, China; tmyan@must.edu.mo (T.Y.); sweetyhukua@sina.com (K.H.); rf870512@163.com (F.R.)

**Keywords:** ginseng, RNA extraction, polysaccharase, polysaccharides, tRNA, post-transcriptional modification

## Abstract

Transfer RNAs (tRNAs) are the most heavily modified RNA species in life entities. Post-transcriptional modifications severely impact the structure and function of tRNAs. To date, hundreds of modifications have been identified in tRNAs, mainly from microorganisms and animals. However, tRNAs in plant roots or tubers that have been widely used for food and medical purpose for centuries are rarely studied because isolation of RNA from plants still remains a challenge. In this paper, a polysaccharase-aided RNA isolation (PARI) method for extraction of high-quality RNA from plants containing large quantities of polysaccharides is developed. This method presents a new strategy of “digesting” polysaccharides that is completely different from the conventional method of “dissolving” the contaminants. By using this method, RNA of high integrity and purity were successfully extracted from ginseng roots because polysaccharide contaminations were removed efficiently with α-amylase digestion. Ginseng tRNAs were first sequenced by NGS and a total of 41 iso acceptors were identified. ChloroplastictRNA^Gly(GCC)^ in ginseng root was purified and four modified nucleosides, including m^7^G, D, T, and Ψ, were identified by LC-MS/MS. The results also revealed that the m^7^G occurs at a novel position 18, which may be related to the deformation of D-loop. PARI is the first enzyme-assisted technique for RNA isolation from plants, which could fundamentally solve the problem of polysaccharide contaminations. By using the PARI method, more individual tRNAs could be isolated easily from polysaccharide-rich plant tissues, which would have a positive impact on the feasibility of research on structure and function of tRNA in plants.

## 1. Introduction

RNA extraction of high integrity and purity is a prerequisite for downstream application. Robust and generic methods for isolating RNA from animal species cells have been well established [1]. However, isolation of high-quality RNA from plant material remains a challenge due either to the degradation of the RNA or a very low yield. This is particularly true for tissues of roots and tubers, which present high levels of polysaccharides, polyphenols, and other secondary metabolites [2,3,4]. TRIzol reagent (Invitrogen; Thermo Fisher Scientific, lnc., Waltham, MA, USA), a typical guanidine isothiocyanate (GITC)-based RNA extraction buffers, is a widely used commercial kit for nucleic acid extraction. TRIzol can remove protein and DNA contamination efficiently, but it cannot remove polysaccharide contamination in starch-rich plant tissues [5,6]. Several cetyltrimethylammonium bromide (CTAB)-based methods have also been developed specifically for the extraction of plant RNA with high levels of polysaccharides [7]. However, even though the CTAB-based method works well for leaf and shoot tissues, isolation of nucleic acids from roots and tubers with dramatically elevated amount of starch and other contaminants still remains a challenge.

Transfer RNAs (tRNAs), a class of highly conserved small non-coding RNA of typically 76 to 90mer, play a crucial role in messenger RNA decoding in nearly all domains of life. In addition to their well-known roles in translation, tRNAs have been shown to perform diverse regulatory functions in other cellular processes, such as apoptosis modulating and stress response programs [8,9]. For a long time, great deal of research efforts have been put into discovering and characterizing the chemical modifications in tRNAs, which have been found to significantly impact structural and functional activities of these biomolecules [10,11]. The most efficient approach for discovering and characterizing tRNA modifications has been based on liquid chromatography tandem mass spectrometry (LC-MS/MS) [12,13,14]. To date, more than 100 modified nucleosides have been identified in 715 tRNAs of 77 different organisms, including mainly bacteria, archaea, animal tissues, and plant leaves [15]. However, tRNAs in plant roots or tubers that have been widely used for food (e.g., potatoes) and medical purpose (e.g., ginseng) for centuries are rarely studied. Fresh leaves are preferred materials for plant tRNA studies, because high-quality tRNA samples can be more easily extracted from leaves than roots and tubers [16,17]. As the epigenetic changes are distinct for different organs and cell types, studies on tRNAs in various organs, including roots and tubers, are imperative [18].

The root of *Panax ginseng* C.A. Mey is a globally well-known traditional Chinese medicine (TCM). Ginseng roots are rich in polysaccharides mainly composed of starch-like glucans and pectins [19], which easily bind or co-precipitate with RNA during isolation by using ethanol, affecting the yield and quality of RNAs. Polysaccharases, such as α-amylase and pectinase commonly used for extraction or structure identification of ginseng polysaccharides [19,20], can hydrolyze polysaccharides efficiently yielding monosaccharides or disaccharides that are more soluble in organic solvents, thus, possibly being useful in removing polysaccharide contaminations from nucleic acid. So far, modification profiles of tRNAs (i.e., identities and locations of modifications in tRNA) in ginseng roots still remain unexplored. In this study, a polysaccharase-aided RNA isolation (PARI) method was developed for the extraction of high-quality RNA from ginseng roots, and the post-transcriptional modifications of tRNA^Gly^ are initially profiled by using a LC-MS/MS method.

## 2. Materials and Methods

### 2.1. Plant Materials 

Fresh roots of *Panax ginseng* C. A. Mey collected from Fusong Town, Jilin, China, were rinsed with distilled water and frozen immediately in liquid nitrogen.

### 2.2. Chemicals and Reagents

TRIzol reagent (Invitrogen), mirVana^TM^ miRNA isolation kit, and SYBR^®^ gold nucleic acid gel stain were purchased from Thermo Fisher Scientific (Waltham, MA, U.S.A.). Cetyltrimethylammonium bromide (CTAB) was purchased from Kingdin Industrial Co., Ltd. (Hong Kong, China). Water-saturated phenol was purchased from Leagene Co., Ltd. (Beijing, China). Chloroform and ethanol were purchased from Anaqua Chemicals Supply Inc. Ltd. (Houston, TX, USA). A-amylase was from *Bacillus subtilis*. Pectinase was from *Aspergillus niger*. Glucoamylase was from Rhizopus and guanidinium thiocyanate was purchased from Tokyo Chemical Industry Co., Ltd. (Tokyo, Japan). A low-rangessRNA ladder was purchased from New England Biolabs (Beverly, MA, USA). A 40% acrylamide/bis solution (19:1), ammonium persulphate (APS), and tetramethylethylenediamine (TEMED) were purchased from Bio-Rad Laboratories Inc. (Hercules, CA, USA). Biotinylated single-stranded DNA oligos (30 mer) were customized from BGI Tech Solutions (Beijing Liuhe) Co., Ltd. (Guangzhou, China).

### 2.3. Extraction of Total RNA from Ginseng Roots

Total RNA were extracted from ginseng roots according to the protocol of the polysaccharases-aided RNA isolation method (PARI), as well as two conventional methods based on CTAB and TRIzol reagent reported previously, with some minor modifications, respectively. To ensure the extraction to be carried out in an RNase-free environment, all surfaces and pipettors were cleaned with RNase Zap solution (Invitrogen; Thermo Fisher Scientific, lnc., Waltham, MA, USA) and plastic wares were autoclaved before use. For the PARI method, around 200 mg ginseng root tissues were ground into a fine powder in liquid nitrogen using a clean mortar and pestle, which was then homogenized in 1 mL TRIzol reagent using a digital dispersing device (IKA, Staufen, Germany). After the plant cells were fully lysed for 10 min at room temperature, an equal volume of chloroform was added to the tissue lysate, followed by vigorous vortex and centrifugation at 12,000× *g* for 15 min at 4 °C. The upper aqueous phase was collected and transferred carefully to a new Eppendorf tube. RNA and polysaccharides in the supernatant were precipitated by adding 1/25 volume of 5 M sodium chloride and 1.25 volume of cold absolute ethanol, and stored at −20 °C for 30 min. After centrifuging at 12,000× *g* for 15 min, polysaccharide contaminations in the pellet were resuspended in water and selectively degraded by polysaccharases, including α-amylase, pectinase, and glucoamylase, respectively, until the pellet was completely dissolved. The hydrolysate was mixed with 2× CTAB buffer (2.0% CTAB, 2% PVP, 2M NaCl, 100 Mm Tris-HCl (pH 8.0), 25 mM EDTA, 1.0% β-mercaptoethanol) and extracted with an equal volume of phenol-chloroform-IAA (50:48:1) by vortexing vigorously. Phases wereseparated by centrifuging at 12,000× *g* for 15 min, and upper phase was collected and extracted again with chloroform-IAA (24:1). The supernatant was collected and mixed with an equal volume of 6 M guanidinium thiocyanate buffer, followed by adding 100% ethanol to a final concentration of 55%. The total RNA in the mixture were isolated and enriched by passing through a filter cartridge containing a silicon gel membrane that immobilizes the RNA. The cartridge was then washed several times with 80% (*v*/*v*) ethanol solution and, finally, the total RNA was eluted with a low ionic-strength solution such as RNase-free water. 

Protocols of CTAB and TRIzol methods modified from Chang et al. [7] and Wang et al. [2] were used in this study. Briefly, after ginseng samples (200 mg) was lysed with CTAB extraction buffer at 65 °C or TRIzol reagent at room temperature, the aqueous phase containing RNA were separated from the organic phase by adding equal volume of chloroform or chloroform-isoamyl alcohol. Total RNA in the supernatant were enriched and recovered by using a filter cartridge of silicon membrane described above in the present of guanidinium thiocyanate and ethanol. 

### 2.4. Quality Assessment of Ginseng total RNA

The yield and purity of the total RNA was spectrometrically assessed by using a NanoDrop spectrophotometer (Thermo Scientific, Waltham, MA, USA) at the wavelengths of 230, 260, and 280 nm. RNA integrity was assessed by its RNA integrity number (RIN) that was automatically generated by an Agilent 2100 BioAnalyzer (Agilent Technologies, Palo Alto, CA, USA). 

The quality of total RNA was also confirmed by RT-PCR. For first-stand cDNA synthesis, 2 µg of total ginseng RNA in a volume of 20 μL was reverse-transcribed following instruction of Transcriptor Universal cDNA Master Kit (Roche). PCR amplifications were performed using primers described previously for cycloartenol synthase gene (Genebank accession number: AB009029) and β-amyrin synthase gene (Genebank accession number: AB014057). The β-actin gene was amplified as the reference gene (Appendix A). Thermocycling was carried out in a final volume of 50 μL containing 5 μL cDNA samples, 1 μL each of the primers (1μM of forward and reverse primers) and 25 μLDreamTaq Green PCR Master Mix (2X) (Thermo Scientific). A thermal cycling profile was conducted as showed in Appendix A. Reaction prodcts were analyzed by electrophoresis on SYBR-stained 2% agarose gels in TAE buffer and visualized with the Gel Doc XR imaging system (Bio-Rad Laboratories Inc., Hercules, CA, USA).

### 2.5. Preparation of Small RNA Fraction and tRNA-enriched Fraction (TEF)

The small RNA species (<200 mer) were isolated from the total RNA of ginseng roots by using a mirVana miRNA isolation kit following the manufacturer’s instruction. tRNAs were then gel-fractionated and purified from small RNA samples by electroelution as described previously with minor modifications [21]. Briefly, 10 μg of small RNA were resolved by 6% urea-PAGE after denaturation for 2 min at 95 °C in loading buffer. After electrophoresis at 200 V in 1× TBE buffer for about 40 min until the bromophenol blue reached the bottom of the gel, RNA were visualized with SYBR^®^ gold nucleic gel stain under UV light. The region of the gel containing total tRNAs was cut out using a clean and sharp scalpel. The band was sliced and the total tRNAs were recovered from the gel by electroelution in dialysis bags of 3000 molecular weight cut-off (MWCO) at 100 V for 90 min in 1× TAE buffer. The eluents in the dialysis bags were collected and the TEF were desalted and concentrated by using the mirVana^TM^ miRNA isolation kit (Ambion; Thermo Fisher Scientific, lnc., Waltham, MA, USA). 

### 2.6. TEF Library Construction and Sequencing

TEF libraries were constructed by using TruSeq small RNA Library Preparation Kit (Illumina, U.S.A.), followed by a round of adaptor ligation, reverse transcription and PCR enrichment (Appendix A). PCR products were then purified and libraries were quantified on the Agilent Bioanalyzer 2100 system (Agilent Technologies, Palo Alto, CA, U.S.A.). The library preparations were sequenced at the Novogene Bioinformatics Institute (Beijing, China) on an Illumina HiSeq platform using the 150 bp paired-end (PE150) strategy. 

### 2.7. Purification of Individual tRNA of Ginseng Roots

Ginseng tRNA^Gly(GCC)^ was isolated from small RNAs by immobilization of the target tRNA onto the streptavidin-coated magnetic beads with specific biotinylated capture DNA probes (Appendix A). Cognate DNA probes, which was designed based on the sequence information from NGS, were incubated with small RNA mixture at 65 °C for about 1.5 h in annealing buffer (1.2 M NaCl, 30 mM HEPES-KOH (pH 7.5), 15 mM EDTA, 0.5 mM DTT). Streptavidin-coated magnetic beads were then added to the mixture and incubated for 30 min at 65 °C. The biotinylated DNA/tRNA coated beads were separated with a magnet for 2 min and washed four times with washing buffer (0.1 M NaCl, 2.5 mM HEPES-KOH (pH 7.5), 1.25 mM EDTA, 0.5 mM DTT). The magnetic beads were re-suspended in RNase-free water and the immobilized tRNA molecules were released by incubation at 70 °C for 5 min. Purified tRNA were analyzed by electrophoresis on a 6% (*v*/*v*) polyacrylamide gels (PAGE) containing 8 M urea prepared according to the manufacturer’s protocol (Bio-Rad Laboratories Inc., Hercules, CA, USA). 

### 2.8. Ribonuclease Digestion of tRNA^Gly(GCC)^

For RNase T1 digestion, 600 ng of tRNA^Gly(GCC)^ was added to 50 U of RNase T1 in 220 mM ammonium acetate buffer, and the mixture was then incubated at 37 °C for 1 h.Pseudouridine in the digestion products were cyanoethylated by reacting with acrylonitrile. The optimal conditions were as follows: 10 μL of RNase T1 digested tRNA was subjected to 26 μL of 41% ethanol/1.1 M triethyl ammonium acetate (pH 8.6) and 4 μL of acrylonitrile, then incubated at 70 °C for 2 h. Digestion samples treated with or without acrylonitrile were condensed for further analysis.

### 2.9. Preparation of Total Nucleosides

A total of 500 ng of tRNA^Gly(GCC)^ was added to 10 μL of 5× enzyme mixture containing 50 U of RNase I, 0.1 U of phosphodiesterase I, 30 U of bacterial alkaline phosphatase, 20 mM MgCl_2_, and 80 mM Tris-HCl (pH 8.0). The reaction system was adjusted to a final volume of 50 μL with RNase-free water, and the mixture was incubated at 37 °C for 3 h. The sample was lyophilized and rehydrated in mobile phase A (0.1% FA in water) described below.

### 2.10. LC-MS Analysis

Intact tRNA and RNase T1 digestion products were analyzed by using a time-of-flight mass spectrometer (6545 QTOF, Agilent) coupled to the 1290 infinity UPLC system equipped with a diode array detector (DAD). Oligonucleotide separations were performed on a Waters Acquity OST C18 column (1.7 µm, 2.1 × 100 mm). Mobile phase A was 15 mM TEA and 100 mM HFIP (pH = 8.5) in MS-grade water, and mobile phase B was a mixture of methanol/water (50:50, *v*/*v*) containing 15 mM TEA and 100 mM HFIP. The chromatographic gradient was shown in Appendix A. Mass spectrums were recorded in negative mode over an m/z range of 600 to 2000 for MS1 scan and 100–1800 for MS2. Collision energy ranging from 10 to 45 V was used for precursor ion fragmentation. System operations and data acquisition were performed by Masshunter Data Acquisition Software (Version B.07.00, Agilent), and the instrument settings were as follows: gas temperature, 320 °C; gas flow, 12 L/min; nebulizer, 35 psi; sheath gas temperature, 350 °C; sheath gas flow, 12 L/min; and fragmenter, 220 V (Appendix A). 

An Agilent 6550 Q-TOF mass spectrometer coupled with 1290 UHPLC was used for identifying isomers of modified nucleosides. The liquid chromatography separation was performed on an Agilent Poroshell 120 HPLC column (2.7 μm, 4.6 × 100 mm) at 35 °C with a mobile phase flow rate of 0.4 mL/min. Nucleotides were eluted using mobile phase A (0.1% formic acid in water) and B (0.1% formic acid in acetonitrile). The gradient started at 1.5% B and increased as follows: 4% B at 4 min, 15% B at 12 min, 25% B at 18 min, then returning to 1.5% B and holding for 3 min. Mass spectra were recorded in positive mode over an m/z range of 100 to 1000 for MS1 scan, and the instrument settings were summarized in Appendix A.

### 2.11. Data Analysis

The tRNA genes were identified by using the tRNAscan-SE 2.0 program (http://lowelab.ucsc.edu/tRNAscan-SE/) and annotated by searching against the Nucleotide Collection (nr/nt) database or *Panax ginseng* chloroplast genome (e.g., AY582139.1 and MH049735.2) using the Basic Local Alignment Search Tool (BLAST) program (https://blast.ncbi.nlm.nih.gov/Blast.cgi). 

Precursor ions of tRNA and digestion products in MS1 data were identified by matching them to their expected m/z values, which were calculated using the MongoOligo on-line calculator (https://mods.rna.albany.edu/masspec/Mongo-Oligo). Sequence informative product ions in LC-MS/MS data of each digestion product were analyzed by RNAModmapper (RAMM) in a variable mode. The mass tolerance was set to 0.8 Da, and *p*-scores of 55 and above were considered to be significant [14].

## 3. Results

### 3.1. Polysaccharase-Aided Extraction of High-Quality RNA from Ginseng Roots

As shown in Figure 1 and Appendix A, the PARI method combined with α-amylase always produced high-quality RNA samples with A260/280 and A260/230 ratios of 1.7–2.1 and RIN beyond 7.4 (in a 1–10 scale), indicating that the RNA were generally pure and intact. Optimal amount of enzyme was also determined and the result showed that adding 50 mg ofα-amylase per 210 mg of ginseng tissue generated high yield (143.2 μg/g) of total RNA with the best spectrometric analysis values (A260/280 = 2.1 and A260/230 = 2.1) and RNA integrity number (RIN = 8). In contrast, extractions with pectinase or glucoamylase produced lower amounts of total RNA, ranged from 18.4 to 121.9 μg/g, with poor purity (A260/230 ranged from 0.2 to 1.5) and integrity (RINs were all below 2.5 mainly caused by low yield).

Subsequently, the PARI method combined with α-amylase was compared with conventional extraction methods including TRIzol and CTAB methods. As shown in Figure 2 and Appendix A, the PARI method employed in sample no. 3 gave the highest RNA yield of 143.2 μg/g ginseng root, which is over 22-fold higher than of TRIzol method (6.4 μg/g in sample no. 16). High yield of RNA product of 115.7 μg/g (no. 17) was also obtained by using CTAB method but with poor integrity. The RNA sample prepared by the PARI method hold a much better RNA integrity number (RIN) of 8.0 than that of CTAB (RIN = 3.1) and TRIzol method (RIN is not available due to low yield). RNA samples thus obtained were spectrophotometrically examined for the possible contaminations. The result showed that PARI methods gave better A260/230 values (ratio = 1.8) than the TRIzol method (ratio = 1.4), suggesting that the PARI method we developed could produce highly pure RNA samples with less contamination of salts, phenol, or polysaccharides. 

To further validate the quality of the resulting RNA, we performed reverse transcription polymerase chain reaction (RT-PCR) with the isolated RNA as the template. Genes encoding cycloartenol synthase and β-amyrin synthase, two critical enzymes for ginsenoside biosynthesis, were amplified together with housekeeping gene β-actin. Figure 3 showed that all three messengers were amplified successfully by PCR and the amplicons migrated as distinct bands at predicted positions (364, 445, and 109 bp for cycloartenol, β-amyrin, and β-actin, respectively) with no signs of degradation, demonstrating that large segments of these mRNA were intact in the original sample. This confirmed that the total RNA extracted from ginseng roots by the PARI method was of satisfactory quality for downstream application.

### 3.2. Ginseng tRNAs Enriching and Sequencing

Figure 4B illustrated that high-quality of small RNA species (<200 mer), consisting mainly of tRNAs (~80 mer) and relatively low amounts of microRNAs (<50 mer) and 5S rRNA (~130 mer), were also prepared by using the PARI method. In contrast, significant degradation of these small RNAs were observed in samples isolated with TRIzol and CTAB methods (Appendix A). Ginseng tRNAs in the small RNA sample were gel-purified (Figure 4B), and this tRNA enriched fraction (TEF) was successfully applied to construct cDNA library for next-generation sequencing (NGS) (Appendix A). The library preparations were sequenced on an Illumina HiSeq platform to generate over 8 million raw paired reads. After removing low quality regions and adaptor sequences, 5772569 clean reads distributing mainly from 60 to 85 mer were obtained, which is consistent with the average length of tRNA species (Figure 4A). From this sequence data, 25 chloroplastic tRNA isoacceptors were identified by searching against the chloroplast genome of *Panax ginseng* reported previously, and 16 new tRNAswere also detected by capturing primary and second structure motif of each sequence in the TEF library (Figure 4C and Appendix A). The relative abundance of each tRNA isoacceptor was estimated by read frequency, and the result showed that tRNA transcripts were highly variable in ginseng root, which might be affected by the codon usage bias [22,23]. It is worth noting that some tRNA modifications represent a fundamental hurdle for the reverse transcriptase during cDNA synthesis. Therefore, certain tRNAs listed in Figure 4C are underrepresented in the cDNA library, which can also contribute to the variations of tRNA read frequencies. 

### 3.3. Profiling of Post-tTranscriptional Modifications of Ginseng tRNA by LC-MS/MS

Ginseng chloroplastictRNA^Gly(GCC)^ was isolated from small RNA by using a solid-phase DNA probe method. As shown in lane4 of Figure 4B, unique band was observed in the predicted position at 74 mer, indicating that the isolated tRNA^Gly(GCC)^ was pure and intact. The high quality of tRNA^Gly(GCC)^ was also apparent from the presence of a single peak with maximum absorption at 260 nm (Figure 5A). The spectrum obtained under gentle ESI interface condition on an Q-TOF platform yielded accurate mass of 23,842.26 Da fortRNA^Gly(GCC)^ (Figure 5B,C), which was 30 Da higher than that calculated from the tRNA gene sequence (23,812.29 Da) obtained by NGS. This suggested that post-transcriptional modifications added mass to ginseng tRNA^Gly(GCC)^, which were taken into account in the final mass spectrograms.

The modified nucleosides in ginseng tRNA^Gly(GCC)^ was then profiled using LC-MS/MS analysis of RNaseT1 digests. As shown in Figure 6, several peaks were observed in the baseline peak chromatograms (BPC) that represented elution of digestion products. Mass spectra corresponding to each peak were analyzed, and the experimentally measured m/z value of each ion was matched to the theoretical value of all unmodified RNase T1 fragments calculated from the gene sequence of tRNA^Gly(GCC)^ (Appendix A). Ion m/z 842.12 and 1685.24 in peak 7 at 7.5 min corresponded to the –2 and –1 charge state of ^14^AAUGGp^19^ + CH_2_ with a theoreticalmass (M_t_) of 1687.05 Da. Location of methyl (CH_2_) was determined by performing collision induced dissociation (CID) on this digestion product. The CID spectrum in Figure 7A showed that over 86%sequence informative product ions were correctly assigned to the theoretical ions of ^14^AAU[mG]Gp^19^ with the highest confidence (*p*-score = 87.20), indicating a methylguanosine (mG) in position 18 of tRNA^Gly(GCC)^ (Figure 8A,C). The structure of mG is assigned as m^7^G rather than other positional isomers (e.g., m^1^G, m2G, and Gm) by matching the retention time of modified nucleoside to that of the corresponding standard substance in the HPLC separation system (Figure 8B).

As shown in the BPC in Figure 6, the triply charged ion *m*/*z* 1579.52 (the corresponding mass = 4741.82 Da) in peak 10 was consistent with the expected values for ^20^UAAAAUUUCUCUUUGp^34^ + 2H (mass = 4739.78 + 2 Da), which indicated a unique mass shift of 2 Da exclusively generated by a dihydrouridine (D). Ionm/z 646.07 and 1293.16 in peak 5 was consistent with the theoretical precursor ions of ^54^UUCGp^57^ + CH_2_ at a charge state of –2 and –1, respectively. In Figure 7B,C, more that 90% product ions in CID mass spectra of both precursor ionsabove were properly assigned with *p*-scores better than 90, whichconfirmed the two modified digestion products as ^20^[D]AAAAUUUCUCUUUGp^34^ and ^54^[mU]UCGp^57^. The structure of mU is assigned as m^5^U based on its retention time (Figure 8B). 

Additionally, eight of the major ions in BPC at *m*/*z* 973.11, 742.62, 996.14, 1020.15, 814.61, 842.12, 980.12, and 1108.63 were found to be consistent with the theoretical m/z values of tRNA gene, suggesting that no modified nucleosides were present in these digestion products (Figure 6 and Appendix A). Sequences of these unmodified products were confirmed by properly assigning at least 70% of the sequence informative product ions with *p*-scores better than 60 (Appendix A).

Pseudouridine (Ψ), a mass-silent modified nucleoside that cannot be detected by MS directly, was distinguished from uridine bycyanoethylation with acrylonitrile. When comparing the MS1 spectra in Figure 6, it is evident that two cyanoethylated digestion products in peaks 5 and 10, ^20^[D]AAAAUUUCUCUUUGp^34^ + C_3_H_3_N (*m*/*z* 1596.85) and ^54^[m^5^U]UCGp^57^ + C_3_H_3_N (*m*/*z* 672.59), give a mass increment of 53 Da compared with the untreated digestion products, indicating that each of them harbors one Ψ. A pseudouridine was assigned to U55 because it is the only nucleoside that could be turned into Ψ in ^54^[m^5^U]UCGp^57^. Positions of Ψ in the other digestion products were determined by LC-MS/MS analysis. The CID mass spectra in Figure 7D pictured the structures as ^20^[D]AAAAUU[Ψ]CUCUUUGp^34^ (*p*-scores = 63.30). At this point, the identity and location of modified nucleosides, including m^7^G, D, T, Ψ, in ginseng tRNA^Gly(GCC)^ were affirmatively identified from the combination of intact mass and sequence confirmation (Figure 8A). 

## 4. Discussion

In this study, polysaccharases were unprecedentedly integrated into the process of RNA extraction of ginseng roots. Three enzymes, including α-amylase, pectinase, and glucoamylase were selected based on the types of ginseng polysaccharides. It can be expected that α-amylase will work better for ginseng roots than other polysaccharases because starch-type polysaccharides readily hydrolyzed by α-amylase make up over 60% of water-soluble polysaccharides of ginseng roots, while substrates for pectinases including various pectins with different pectic domains account for only a small portion (<9%) of ginseng’s polysaccharides (Appendix A) [19]. Actually, α-amylase made a real difference to ginseng RNA isolation in our lab. As shown in the left tube (No. 1) in Figure 1F, large amounts of polysaccharides released from ginseng tissues clumped up in ethanol-water solution to form a sticky and chunky wad wrapping almost all the nucleic acid up in it. When α-amylase was added, these polysaccharides were rapidly and specifically hydrolyzed to monoses that are freely soluble in high concentration of ethanol (tube no. 2 in Figure 1F), keeping the RNA free from contamination and degradation. 

Compared with TRIzol and CTAB method, the PARI method is better at the isolation of high-quality RNA from polysaccharide-rich planttissues. The low yield problem of the TRIzol method is due mainly to the co-precipitation of a large portion of ginseng RNA with polysaccharides that were difficult to dissolve when they were exposed to a high concentration (>30%, *v*/*v*) of alcohol. CTAB naturally holds the power to segregate RNA from polysaccharide. However, severe degradation of ginseng RNA in CTAB extraction buffer was observed in this study (Figure 2). This is mainly caused by heating the lysis buffer to 65 °C at the first step of the CTAB protocol [7], which is necessary and inevitable for releasing nucleic acid out of cells completely. Although several modified CTAB methods with heating have been developed for many plant species, none of them could be applied successfully in our lab for the extraction of ginseng RNA because they could be more susceptible to degradation by heating even within two minutes. In the PARI method, the ginseng root was treated at room temperature, thus avoiding degradation of RNA.

The PARI method provides a reliable and efficient way to collect high purity and integrity of RNA from ginseng roots, which is critical for the overall success of downstream application. In our lab, the PARI method can successfully be up-scaled to extract hundreds of milligrams of total RNA from various plant tissues including ginseng roots, ligustrum fruits and notoginseng roots, which further illustrate the versatility of the method. This protocol is supposed to be universally applicable to isolation of not only RNA but also DNA from other plant tissues rich in polysaccharides, no matter what kind and how many of them, with reasonable amounts of proper polysaccharases. 

tRNA sequencing is important for structural and functional study of tRNAs, because nearly half of tRNA genes remain silent in life entitiesand the genome data cannot describe the practical transcriptional profile of tRNAs in cells [25]. So far, tRNAs in ginseng root have never been sequenced and little genomic information is available. In this study, ginseng tRNAs were first sequenced by NGS, because high-quality tRNA samples were prepared by using the PARI method. These tRNA-seq results initially confirmed the successful transcription of chloroplast tRNAs in ginseng root, and additionally provided some new tRNA sequence information besides chloroplast tRNAs, which could be helpful for ginseng tRNA studies such as structure and modification characterizations. These new ginseng tRNAs were found to be highly homologous with cytoplasmic or mitochondrial tRNAs of other plant species (e.g., *Arabidopsis thaliana* and *Ilex pubescens*), which could be attribute to the high evolutionary conservation of tRNAs across plant species [26].

It is a well-known fact that synonymous codons are not used with equal frequencies, which shape the tRNAomes in both eukaryotes and prokaryotes. A statistical analysis of the chloroplast genome in *Panax ginseng* based on codon count (relative synonymous codon usage, RSCU) indicated that 31 codons in ginseng chloroplast genome with RSCU larger than 1.0 were preferred [22]. However, we surprisingly found that many corresponding anticodons were obviously completely absent in the genome (KF431956.1) or minimally transcribed (Figure 4C), suggesting a low correlation between the practical codon usage bias and the codon frequencies in genomes. On the other hand, there is evidence of a strong positive relationship between codon usage bias and tRNA abundance, which may partially explain the variability of tRNA transcripts [23]. However, this correlation was not observed in *Panax ginseng*, mainly because of a fact that certain tRNAs with heavy modifications could be significantly underrepresented during the NGS process. 

In the present study, primary structure with post-transcriptional modifications of chloroplastictRNA^Gly(GCC)^ in ginseng root was characterized by using a LC-MS/MS method. Since the tRNA genehad been confirmed by NGS, correlations between modified sequence and CID product ions became conceptually straightforward so that we could map the modification onto the tRNAs with high confidence. Two contiguous modifications, m^5^U-Ψ, were identified at positions 54 and 55, respectively, in the T-loop of tRNA^Gly(GCC)^ (Figure 8A), which were consistent with the conserved location of TΨ in thecanonical cloverleaf structure [24,27]. An atypical D-stem of only 2bp was observed in tRNA^Gly(GCC)^, resulting in a larger D-loop that, as expected, contained a D at position 20.This result parallels published findings that D commonly occurs at position 16, 17, and 20,which flank the two highly conserved guanines at positions 18 and 19 (Figure 8) [24,28]. Unexpectedly, an m^7^G was also found in D-loop at a novel position 18. The m^7^G is catalyzed by tRNA (guanine-*N7*-)-methyltransferase commonly at position 46 guanine in a variable loop, which is critically dependent on the structure motif of a loop (the variable loop) inserting into two stems (the anticodon-stem and T-stem) [29,30]. Herein, due to the truncated D-stem, a similar motif was formed in the D-loop region combined with anticodon- and acceptor-stem that fits in the methyltransferase, which was supposed to facilitate the *N7*-methylation of guanine at position 18. 

Wobble position 34 in anticodon region and position 37 are two most frequently modified tRNA residues that are essential for translational efficiency and fidelity. However, these modifications are completely absent in ginseng tRNA^Gly(GCC)^ (Figure 8). tRNAs lacking modifications have been found in many other species, especially in organelles and single cell organisms with small genomes. It has been reported that tRNA^Gly(GCC)^ without any base modifications in anticodon regions mainly participates in transcriptional regulation and transcription. Instead, the tRNA^Gly(UCC)^, which was also identified in ginseng chloroplasts (Appendix A), more often participates in translation by decoding all four glycine codons by “superwobbling” [31].

## Figures and Tables

**Figure 1 biomolecules-10-00621-f001:**
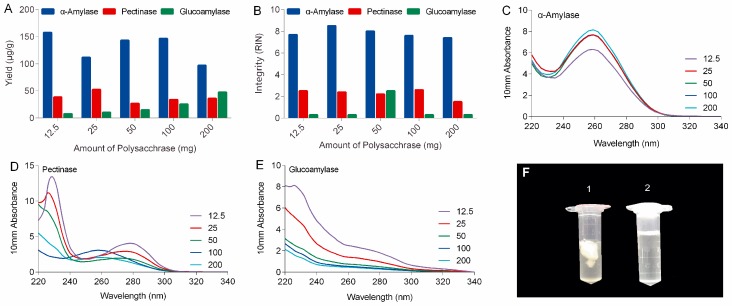
Comparison of the yield (**A**), integrity (**B**), and purity (Nanodrop spectrophotometric measurements in **C**–**E**) of total RNA isolated from ginseng roots using PARI method with different amount of α-amylase, pectinase and glucoamylase. (**F**) Comparison of precipitation before (no. 1) and after (no. 2) α-amylase digestion.

**Figure 2 biomolecules-10-00621-f002:**
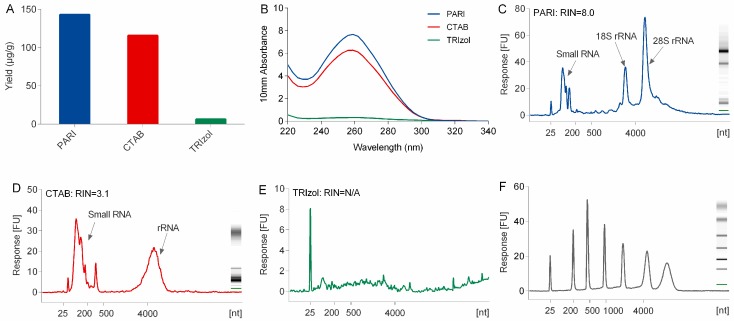
Comparison of the yield (**A**), purity (**B**), and integrity (RIN values reported by bioanalyzer in **C**–**E**) of ginseng RNA isolated using the PARI mehthod, TRIzol reagent, and CTAB method. (**F**) RNA ladder (25–6000 mer).

**Figure 3 biomolecules-10-00621-f003:**
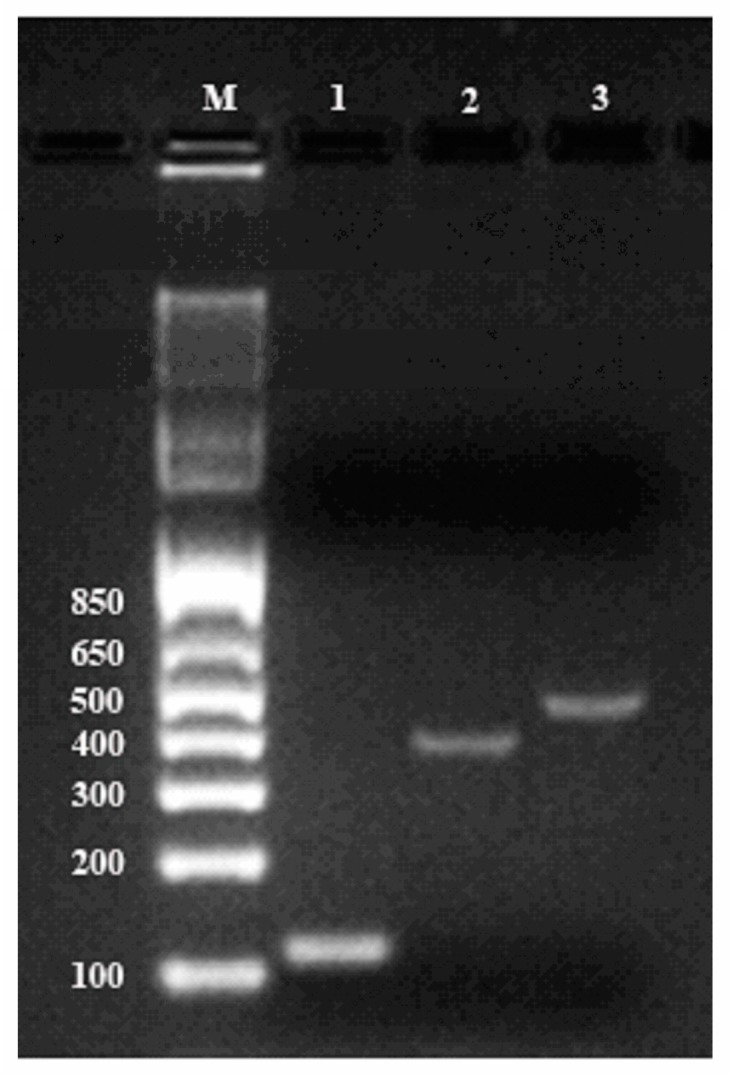
Verification of PCR products on 2% agarose gel electrophoresis along with the 1 Kb Plus DNA Ladder (M). Amplification of each mRNA is shown: lane 1, β-actin (109 bp); lane 2, cycloartenol synthase (364 bp); lane 3, β-amyrin synthase (445 bp).

**Figure 4 biomolecules-10-00621-f004:**
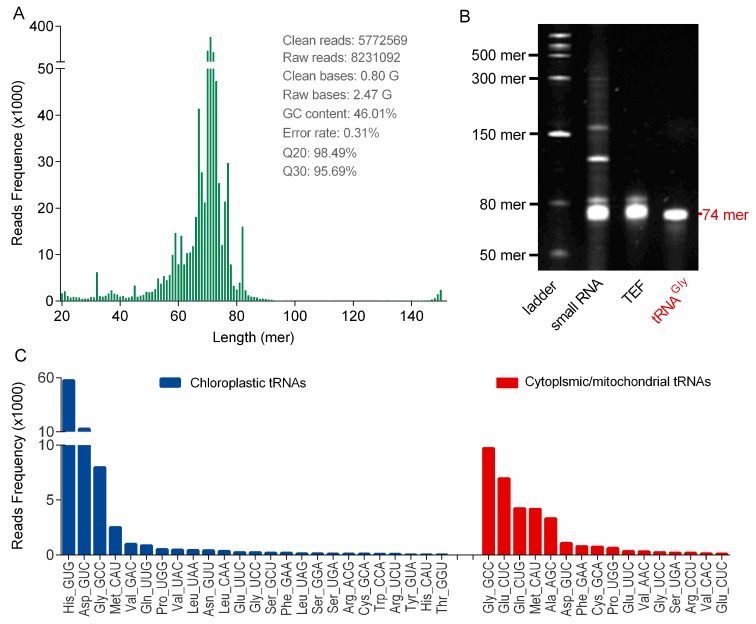
(**A**) Read length distribution of 5,772,569 clean reads produced from the library of ginseng tRNA enriched fraction (TEF) by NGS. (**B**) Denaturing PAGE analysis of small RNA of ginseng roots (lane 2), TEF (lane 3), and tRNA^Gly(GCC)^ (lane 4). RNA ladder ranged from 50 to 1000 mer is displayed in lane 1. (**C**) Ginseng tRNA isoacceptors identified by NGS. The chloroplastic tRNA isoacceptors were shown in blue bars and the new transcriptionally silent tRNA genes were highlighted in red. The relative abundance of tRNA transcripts were measured by read frequency.

**Figure 5 biomolecules-10-00621-f005:**
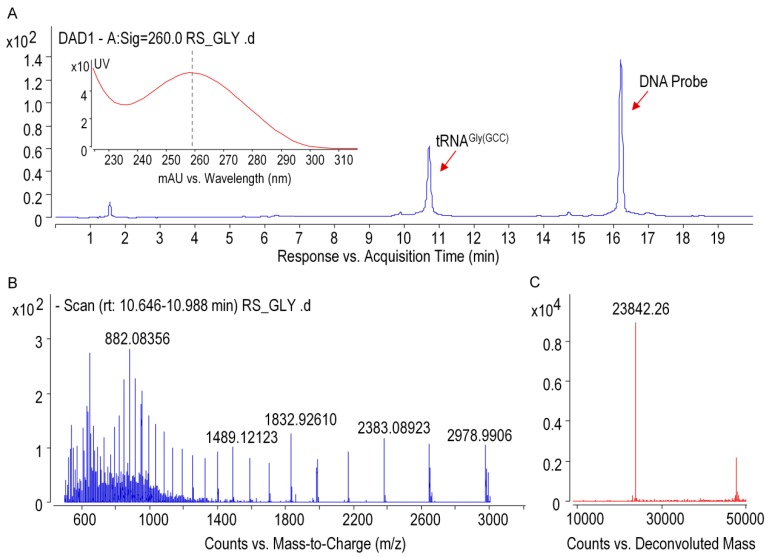
Qualification of purified tRNA^Gly(GCC)^ by using mass spectrometer. (**A**) UHPLC-DAD chromatogram (the inset shows UV-absorption spectrum of tRNA), (**B**) ESI-MS spectra of tRNA^Gly(GCC)^ with multiple negative charges, and (**C**) deconvoluted MS spectra of tRNA^Gly(GCC)^.

**Figure 6 biomolecules-10-00621-f006:**
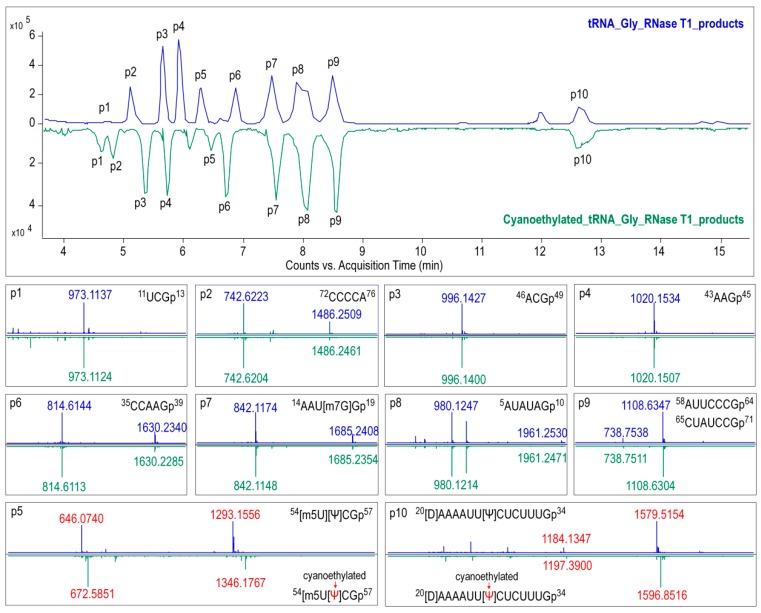
Baseline peak chromatograms (BPC) from LC-MS/MS analysis of RNase T1 digests of tRNA^Gly(GCC)^ treated with (green) or without (blue) acrylonitrile. Mass spectra corresponding to each peak (p1 to p10) are extracted and displayed below the BPC. The red tag indicates the precursor ions with mass shift (of 53.027 for single charged ions) corresponding to the presence of cyanoethylated pseudouridine (Ψ) in the digestion products.

**Figure 7 biomolecules-10-00621-f007:**
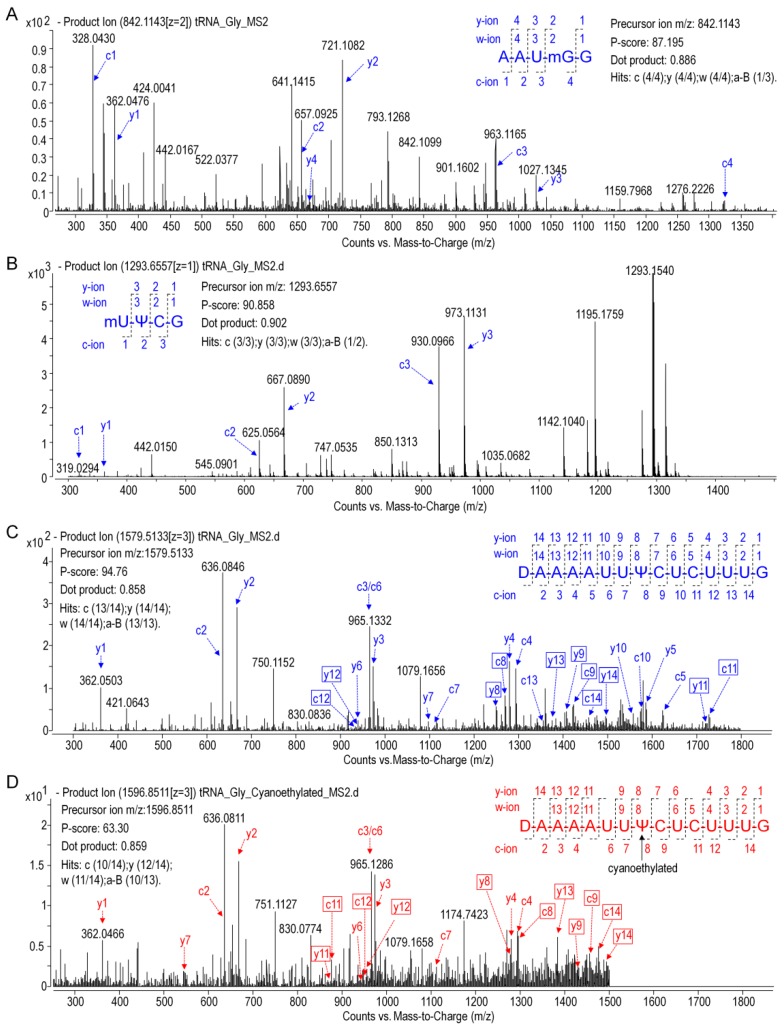
CID mass spectra of digestion products with precursor m/z 842.1143 (**A**), 646.0732 (**B**), and 1579.5131 (**C**). (**D**) CID spectra of cyanoethylated product with precursor *m*/*z* 1596.8511. Product ions are properly assigned and scored by RAMM program and the representative c and y ions are labeled and plotted. Unique product ions of both cyanoethylated and uncyanoethylated products are highlighted in boxes.

**Figure 8 biomolecules-10-00621-f008:**
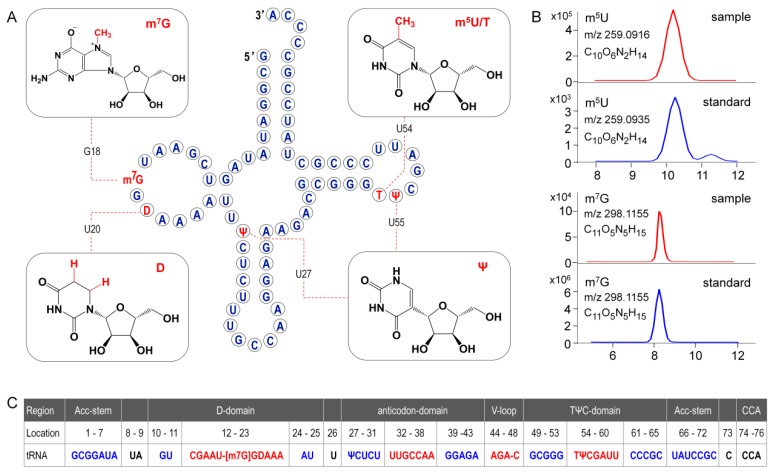
(**A**)Secondary structure of ginseng chloroplastictRNA^Gly(GCC)^. Modified nucleosides are indicated in red. (**B**) Position isomers of monomethylated G and U are identified based on their retention times in standard UHPLC separation system. (**C**) Formating and numbering of nucleotides in tRNA^Gly(GCC)^ have been done in accordance with Sprinzl et al. [24]. Stems are highlighted in blue and loops are indicated in red. Positions 17 and 47, which are empty, are indicated by dashes.

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
