# Peer review of "LC-MS/MS Profiling of Post-Transcriptional Modifications in Ginseng tRNA Purified by a Polysaccharase-Aided Extraction Method"

_biomolecules, 2020, doi:10.3390/biom10040621_

Round 1

Reviewer 1 Report

This manuscript presents a method for purifying tRNA from polysaccharide-rich plant sources. The method relies on enzymatic digestion of the polysaccharides and is shown to be more effective in removing polysaccharide contaminants than the CTAB method typically used for plant tRNA preparation, and the TRIzol method used for general RNA extraction. The authors also show that tRNA prepared in this manner can be successfully sequenced by NGS and profiled for tRNA modifications by LC-MS/MS. The data is substantial, well presented, and show that alpha-amylase digestion works best in producing high quantities of intact and polysaccharide-free tRNA. The study also uncovers new apparently transcriptionally silent tRNA genes in ginseng chloroplast and identifies a new location of an N7-methlated guanine in tRNA. Overall, the study provides valuable methods for plant tRNA research especially in the tRNA modification field, and establishes a more complete tRNA set for ginseng root.

Some points:

The relative abundance of tRNAs are reported (Fig. 4) but an unrelated paper about amino acid usage in bacteria is cited (ref 22). I wonder if the authors can instead comment on how these relative abundances compare with codon usage in chloroplast of Panax ginseng or related species (e.g. doi: 10.11648/j.ijhnm.20170305.11).

In Table S10, I assume the color code in the first column indicates the 16 tRNAs that are transcriptionally silent, but it helps to indicate that in the table, and in Fig. 4C.

Correction of grammatical and typographical errors throughout the text and supplementary tables is necessary. A few examples (but many more need attention):

Line 30: “tRNA individuals”.. individual tRNAs

Line 59: diverse

Line 133: “Separated phases by …”  Phases were separated by …

Expand the acronym CID

Lines 454 & 456: “polysaccharases” polysaccharides.

It is not clear what is intended by the statement at lines 517-520.

Line 533: “position 17”  18.

Reviewer 2 Report

Yan et al. submitted a manuscript describing an RNA extraction method from ginseng roots that is superior compared to previously described protocols for other plant tissues. The main point of the new purification approach is the use of alpha-amylase to remove the high polysaccharide content present in roots. The quality of the extracted RNA was tested and the small RNA population (< 200 nt) was deep sequenced. This allowed the authors to identify 16 new tRNAs in ginseng. tRNA-Gly(GCC) was affinity purified and its post-transcriptional modification pattern analyzed by LC-MS/MS. This demonstrates that the described RNA extraction procedure is compatible with downstream applications thus highlighting the relevance of the method for other plant RNA biologists.

The manuscript is well written and reports on a straightforward procedure to isolate and purify total RNA from roots. Nevertheless, several points merit consideration:

1) The manuscript would gain significance if the authors could demonstrate that the purification methods would also be superior in other plant species and tissues. This is especially important to demonstrate since ginseng is definitely not among the main plant model systems in use in the field.

2) Page 9, line 312: maybe change to “…three messengers were amplified successfully by PCR…”. The word “processed” sounds funky in that context.

3) Fig. 3: to unequivocally demonstrate the quality and intactness of the RNome after their extraction method, showing an ethidium bromide stained gel of the total RNA extract would be beneficial. Only such a direct visualization of the quality of the isolated RNA would convince readers. The RT-PCR results shown in Fig. 3 are clean, but do not report how the initial RNA templates actually looked like (e.g how degraded the total RNA sample actually was).

4) Fig. 3; figure legend: Please rephrase. e.g.: “Amplification of each mRNA (not gene) is shown: lane 1…..)

5) Page 10: lanes 339-344: under the applied experimental conditions RNA-Seq is not quantitative when it comes to tRNA reads. It is a well-known fact that several tRNA modifications represent a fundamental hurdle for the reverse transcriptase during cDNA generation. Therefore certain tRNAs are underrepresented in the cDNA library and based on that one cannot draw any quantitative conclusion. This needs to be taken into consideration here.

6) There seems to be a nomenclature problem. Two examples: In the text the authors refer to m7G or dihydro-U being located at positions 18 and 20, respectively. However in Fig. 8A these two (and others) modifications are shown at positions G17 and D20! On page 18, lane 533: this m7G is described as being at position 17.

7) The two post frequently modified tRNA positions are residues 34 and 37 in the anticodon loop. The authors need to discuss the obvious complete absence of these modifications in ginseng tRNA-Gly(GCC).

8) Page 18, lines 522-540: this paragraph is identical to the abstract and should be removed.
